# Construction of Patterned Cu_2_O Photonic Crystals on Textile Substrates for Environmental Dyeing with Excellent Antibacterial Properties

**DOI:** 10.3390/nano14181478

**Published:** 2024-09-11

**Authors:** Zhen Yin, Chunxing Zhou, Yiqin Shao, Zhan Sun, Guocheng Zhu, Parpiev Khabibulla

**Affiliations:** 1College of Textile Science and Engineering, Zhejiang Sci-Tech University, Hangzhou 310018, China; 2023220202127@mails.zstu.edu.cn (Z.Y.); 2023210201085@mails.zstu.edu.cn (C.Z.); syq@zstu.edu.cn (Y.S.); 2021327100055@mails.zstu.edu.cn (Z.S.); 2Zhejiang-Czech Joint Laboratory of Advanced Fiber Materials, Zhejiang Sci-Tech University, Hangzhou 310018, China; 3Department of Technology of Textile Industry Products, Namangan Institute of Engineering and Technology, 7, Kasansay Street, Namangan 160115, Uzbekistan; parhabib@mail.ru

**Keywords:** textile substrate, structural color, photonic crystals, Cu_2_O microspheres, high refractive index

## Abstract

Structural dyeing has attracted much attention due to its advantages such as environmental friendliness, vivid color, and resistance to fading. Herein, we propose an alternative strategy for fabric coloring based on Cu_2_O microspheres. The strong Mie scattering effect of Cu_2_O microspheres enables the creation of vibrant structural colors on fabric surfaces. These colors are visually striking and can potentially be adjusted by tuning the diameter of the microspheres. Importantly, the Cu_2_O spheres were firmly bonded to the fabrics by using the industrial adhesive PDMS, and the Cu_2_O structural color fabrics exhibited excellent color fastness to washing, rubbing, and bending. Cu_2_O structural color fabrics also demonstrated excellent antimicrobial properties against bacteria such as *Escherichia coli* (*E. coli*) and *Staphylococcus aureus* (*S. aureus*). The bactericidal rates of Cu_2_O structural color textiles after washing for *E. coli* and *S. aureus* reached 92.40% and 94.53%, respectively. This innovative approach not only addresses environmental concerns associated with traditional dyeing processes but also enhances fabric properties by introducing vibrant structural colors and antimicrobial functionality.

## 1. Introduction

To date, the primary method of fabric coloring involves attaching colored natural or synthetic dyes to fabric fibers. These dyes rely on chemical chromophores to absorb specific wavelengths of light, which allows the human eye to perceive color [1]. These colors are classified as pigmented colors. In contrast, structural colors represent another method of generating color. They arise from interactions such as interference, diffraction, and scattering that occur on nanostructure surfaces, comparable to the wavelengths of incident light and light spectra [2,3]. Due to the unique properties of these nanostructures, it is possible to achieve specific functionalities such as antimicrobial properties, hydrophobicity, self-cleaning capabilities, and UV resistance, and for them to act as a photonic biosensor to detect viruses [4,5,6]. Compared to conventional fabric dyeing technologies, structural colors offer significant advantages such as being eco-friendly, vibrant in color, and resistant to fading. In the context of advocating for green and sustainable practices, structural colors hold considerable promise. The dyeing process for coloring fabrics is maturing, mainly including dip-dyeing, tie-dyeing, cold-dyeing, printing, and dyeing, and the products obtained can cover all aspects of people’s lives. However, there are still some problems against the background of advocating green and sustainable development: (1) the dyeing process requires dyes, various chemicals, and additives; processing is cumbersome and complex; and energy consumption and water consumption are high; (2) each process generates large volumes of colored wastewater, which is difficult to treat; the wastewater composition is complex, with high treatment costs that are difficult to effectively lower; and (3) it is easy to pollute the environment. The structure of the pigment molecule itself is complicated, containing several aromatic rings bonded into a fully conjugated system with a long series of alternating single and double bonds between carbon atoms and other atoms, which make it difficult to synthesize artificially. In addition, the structure of the pigment molecule itself is not stable, and it is easily decomposed under long-term sunshine [7,8,9]. If the structural color is applied to fabric coloring, it is hoped that this will fundamentally solve the problems of conventional fabric coloring and offer a new textile coloring method [9,10,11].

In recent years, Shao et al. have developed a method to create periodic layers of PET films and SiO_2_ films on polyester fabric surfaces. By varying the particle size or the number of SiO_2_ nanoparticle cycles, they achieved polyester fabrics in different colors. Moreover, these fabrics exhibit iridescence, displaying distinct rainbow colors at different angles [12]. Lee et al. have also made significant strides by fabricating conductive e-textiles and color-coated e-textiles. They achieved this by depositing multilayers of Al_2_O_3_/TiO_2_ on conductive e-textiles. By adjusting the thickness of the Al_2_O_3_ and TiO_2_ layers, they could change the color of the coated conductive e-textiles to purple, green, or pink, demonstrating versatile color control through layer thickness manipulation [13,14,15].

Compared to one-dimensional and two-dimensional photonic crystals, three-dimensional photonic crystals exhibit photonic band gaps in all directions, which effectively block the propagation of light at specific frequencies and incident angles. This property has spurred on intensive research into three-dimensional photonic crystals for fabric coloring applications [16,17]. To facilitate the assembly of highly ordered structures, low-refractive-index nanorods (with *n* < 1.6), such as PS, PMMA, P(St-MAA), and SiO_2_, are commonly used as the building blocks. These nanorods are easy to synthesize and offer high uniformity. However, constructing such structures on fabric surfaces presents challenges due to the inherent complexity of intertwined yarns that make up the fabric structure [17,18]. Liu et al. have made significant progress in this area by successfully assembling photonic crystals on polyester fabric surfaces using SiO_2_ microspheres and P(St-MAA) microspheres. This approach has enabled the creation of brightly colored polyester fabrics. Their research has emphasized the crucial relationship between the fabric substrate and the photonic crystal structure. Specifically, they found that dense and flat fabric substrates facilitate the formation of ordered photonic crystals, laying a solid foundation for integrating photonic crystals into fabrics [19,20,21]. In summary, the application of three-dimensional photonic crystals to fabric coloring holds promise due to their ability to produce vibrant colors and their potential for creating functional textiles. Overcoming challenges in fabric substrate compatibility and structural complexity remains a focus of ongoing research in this field [22,23].

Fabrics created using photonic crystal structures exhibit traits such as high brightness, intense saturation, and iridescent colors [24]. However, photonic crystals face challenges including stringent structural requirements (requiring highly ordered structures), prolonged assembly times, and color variability with viewing angle changes. Additionally, issues like poor colorfastness due to microsphere point contact and reduced breathability further restrict their use in textile coloring. These factors have constrained the advancement of photonic crystals for textile applications [25,26]. The development of textile coloring has been limited by various challenges. Recently, Ge et al. successfully addressed some of these issues by leveraging the capillary effect of fabric substrates. They adopted a spraying method to precisely assemble SiO_2_ micelle photonic crystals on the surfaces of single-cluster fibers. This approach is not only easier and quicker but also avoids the clogging of fabric weaving holes, which had been problematic in previous methods [27]. Furthermore, they applied a protective binder layer over the assembled structure to enhance its stability. As a result, they achieved fabrics with saturated colors, solid dyeing, soft textures, and excellent breathability, complemented by an iridescent appearance. These advancements highlight a promising pathway for the large-scale production of high-quality structural color fabrics [28,29,30,31].

In this study, Cu_2_O microspheres of varying diameters were synthesized by adjusting the ratio of copper acetate to trisodium citrate. These microspheres were then utilized as coloring agents to create structural colored fabrics using a spraying method [32,33]. This study investigated the color fastness of the Cu_2_O structural colored fabrics under various conditions. Additionally, the antibacterial properties of the Cu_2_O structural colored fabrics were investigated by assessing their effectiveness against both Gram-positive and Gram-negative bacteria. Overall, this research aimed to establish an experimental foundation for developing functional structural colored fabrics characterized by vivid colors and strong antibacterial properties.

## 2. Materials and Methods

### 2.1. Materials

Copper acetate monohydrate (C_4_H_6_CuO_4_·H_2_O), sodium citrate dehydrate (C_6_H_5_Na_3_O_7_·2H_2_O), polyvinylpyrrolidone, ethylene glycol, ascorbic acid, sodium hydroxide, and anhydrous ethanol were provided by Shanghai Macklin Biochemical Technology Co., Ltd. (Shanghai, China). Deionized water (conductivity 18 MΩ·cm) was prepared in the laboratory. The woven polyester fabrics were obtained from Guangdong Yuluo Textile Co., Ltd. (Guangzhou, China). SYLGAR 184 Silicone Elastomer (PDMS) was purchased from Dow Corning Co., Ltd. (Midland, MI, USA). All the chemicals and solvents were of the analytical grade and were used as received without further purification.

### 2.2. Preparation of Cu_2_O Microspheres

In this experiment, a two-step liquid phase reduction method was used to synthesize Cu_2_O microspheres, and the preparation of 210 nm Cu_2_O microspheres was used as an example, with the following steps.

Firstly, the copper precursor solution was prepared: 1.5 g of PVP (Mw = 55,000), 0.52 g of trisodium citrate dihydrate (C_6_H_5_Na_3_O_7_·2H_2_O), and 0.33 g of copper acetate monohydrate (C_4_H_6_CuO_4_·H_2_O) were weighed into a beaker, and then 30 mL of glycol and 100 mL of deionized water were added to the beaker, with magnetic stirring for 20 min and ultrasonic shaking for 20 min, and finally, the solid powder was completely dissolved to form a clear and transparent light blue solution.

Subsequently, we synthesized Cu_2_O microspheres: the copper source precursor solution was transferred to a 500 mL beaker, under the condition of a 20 °C water bath. Then, 20 mL of 0.4 M sodium hydroxide solution was slowly dripped into the precursor solution, and at this time, the solution became dark blue. We stirred vigorously for 10 min and then added in two steps 15 mL of 0.2 M ascorbic acid reducer. The reaction was stopped after 60 min of stirring and centrifuged at 8000 r/min for 10 min to obtain the reaction product, which was washed once with water and once with ethanol to remove impurities, and dried under vacuuming. By varying the ratios of copper acetate and trisodium citrate (0.8, 1.0, 1.1, and 1.2), Cu_2_O nano-microspheres with corresponding particle sizes of 165 nm, 210 nm, 240 nm, and 275 nm were finally prepared. The diameters of Cu_2_O microspheres prepared in different ratios were different, with a few nanometers of deviation in the particle size. The preparation process is shown in Figure 1.

### 2.3. Preparation of Cu_2_O Structural Colored Fabric

In this experiment, the process used to prepare structural colored fabric involved several steps: Firstly, we mixed 184 silicone elastomer base material and 184 silicone elastomer curing agent in a 10:1 ratio and stirred the mixture with suitable deionized water for 2 h to achieve a 20% PDMS (polydimethylsiloxane) solution. Secondly, the synthesized Cu_2_O microspheres were dispersed in anhydrous ethanol and ultrasonicated for 20 min to obtain a uniformly dispersed Cu_2_O microsphere dispersion (2 wt.%). A mixed solution of PDMS and Cu_2_O was sprayed onto the surface of polyester fabric using a spray gun connected to an air compressor. The compressor pressure was set to 0.5 PSI, and the spray gun nozzle diameter was set to 0.2 mm. The Cu_2_O-PDMS coated fabric was dried at 50 °C for 3 h to obtain the structural colored fabric.

### 2.4. Characterization Methods

The surface morphology of the Cu_2_O microspheres was determined using a field emission scanning electron microscope (FESEM, Gemini500, Oberkochen, Germany). The elemental composition of the Cu_2_O microspheres was measured by energy dispersive X-ray spectroscopy (EDS) using a field emission scanning electron microsphere (FESEM, Gemini 500, Oberkochen Germany). The crystal phase structures of the Cu_2_O microspheres were analyzed by an X-ray diffractometer (XRD, K-alpha, Thermo Fisher, Waltham, MA, USA). The dried Cu_2_O microspheres were ground into powder as a sample, where the emission source was CuKα ray (wavelength was 0.154178 nm), the scanning speed was 10°/min, the scanning angle was 10°~80°, and the electric current and electric voltage were 40 mA and 40 kV, respectively. The reflection spectra of the structural colored fabrics were recorded using an Ocean Optics fiber optic spectrometer (Maya 2000, Dunedin, FL, USA) and a spectrophotometer (Hitachi U-4100, Tokyo, Japan). The optical images of the structural colored fabrics were acquired using a 3D video microscope (HIROX KH-7700, Tokyo, Japan).

### 2.5. Characterization of Structural Color and Its Color Fastness

Bending test: The reduction in structural color and shedding of photonic crystals were observed before and after the bending test. Washing fastness: We placed the fabric in a 150 mL beaker filled with water and magnetically stirred it at 1200 r/min for 15 min. Friction test: A friction fastness test was performed according to the national standard ISO 105-X12:2001 [34]. Light resistance test: A light resistance test was performed according to GB/T 8427-2019 [35].

### 2.6. Antimicrobial Testing

Antimicrobial performance: According to GB/T 20944.3-2008 “Evaluation of Antimicrobial Performance of Textiles Part 3: Oscillation Method”, *E. coli* and *S. aureus* are suitable test strains [36]. The bacteria were purchased from Beijing SanYao Science & Technology Co., Ltd. (Beijing, China). The specific test procedure was as follows:

The samples to be examined were cut into pieces of 5 mm × 5 mm size, weighed at 0.75 g, and sterilized by pressure steaming at 103 kPa and 125 °C for use. The control sample was treated in the same way. Triangular vials were filled with the control sample and treated sample, and then 70 mL of 0.03 mol/L phosphate (PBS) buffer was added to each triangular vial. Next, we added 5 mL of bacterial suspension to the above triangular vials and placed them on a thermostatic shaker at 150 r/min and 25 °C for 18 h. After 18 h of contact shaking, 1 mL of liquid was extracted from each flask and added to 9 mL of 0.03 mol/L PBS buffer. The solution was thoroughly mixed and diluted to the appropriate dilution gradient using a 10-fold dilution method. Then, 0.1 mL was extracted from each dilution gradient and added to agar medium. Two parallel samples were prepared for each dilution and then inverted after coating evenly, and then they were continuously cultured at 37 °C for 24 h in a thermostatic incubator. The sample was incubated at 37 °C for 24 h, and the number of colonies on each medium was photographed, counted and recorded.

We calculated the inhibition rate of the sample to be tested against the target strain:(1)$$Y=\frac{W_{b}-W_{c}}{W_{b}}\times 100\%$$
*Y*—inhibition rate of the sample to be tested (%);*W_b_*—concentration of live bacteria after 18 h of the blank sample (CFU/mL);*W_c_*—concentration of live bacteria after 18 h of the sample to be tested (CFU/mL).

## 3. Results and Discussion

### 3.1. SEM Images of Cu_2_O Structural Colored Fabric

Scanning electron microscopy (SEM) was used to observe the morphology of the fabric surface after spray-coating Cu_2_O, as shown in Figure 2. Cu_2_O microspheres with four different sizes were synthesized from different conditions, which were 165 nm, 210 nm, 240 nm, and 275 nm. The photo in the upper right corner corresponds to Cu_2_O microspheres of different particle sizes exhibiting different colors in a Petri dish. The Cu_2_O microspheres have smooth surfaces, excellent sphericity, and a uniform size.

The SEM image in Figure 3A is the surface fiber of the fabric, and it can be observed that Cu_2_O microspheres are evenly distributed on the surface of the fiber. After the SEM image is enlarged, as shown in Figure 3B, it can be observed that the surface microsphere particles have a uniform particle diameter. The prepared Cu_2_O microspheres were characterized by EDS, and the results are shown in Figure 3C. As shown in Figure 3C, the spectrum contains only copper and oxygen elements. Energy spectrum analysis of each element shows multiple peaks, with the number of peaks determined by the number of electron layers of the element. The more electron layers an element has, the more peaks it shows. Therefore, the copper element has a two-peak spectrum. As is shown in the Table 1, the table of elemental composition, the element Cu occupies 89.50% and the element O occupies 10.50%, and the sum of the two is 100.00%. Therefore, the EDS analysis results show that the prepared microspheres are Cu_2_O single-crystal microspheres.

### 3.2. XRD Analysis and Size Distribution of Cu_2_O Microspheres

The crystal structure of the prepared Cu_2_O microspheres was characterized using XRD, and the results are shown in Figure 4A. As shown in Figure 4A, the prepared microspheres showed characteristic diffraction peaks at 2θ = 29.6°, 36.5°, 42.4°, 61.4°, 73.6°, and 77.4°, corresponding to (110), (111), (200), (220), (311), and (222) crystal planes, respectively. The XRD spectra of cuprous oxide microspheres are consistent with the standard cubic cuprous oxide spectra in PDF No. 05-0667, and no other phases such as copper oxide are present in the XRD spectra. Therefore, the XRD spectra show that the prepared microspheres are Cu_2_O single-crystal microspheres.

Using a particle size meter, the diameter distribution of Cu_2_O microspheres was measured. As shown in Figure 4B, the results showed that the particle size distribution was concentrated at 165 nm–275 nm and the microspheres were well prepared.

### 3.3. Color Properties of Cu_2_O Structural Colored Fabric

Due to the photonic forbidden bands present in the photonic crystal itself, light at a specific wavelength cannot propagate and is reflected back, resulting in interference diffraction in an arrangement of periodic media. If the wavelength is in the visible region, the human eye will be able to see the vibrant structural colors. The principle of color-generating structures in photonic crystals conforms to Bragg’s law of diffraction. It can be expressed as Equation (2):(2)$${m\lambda }_{max}=2d_{hkl}{\left(n_{avg}^{2}-{sin}^{2}\theta \right)}^{1/2}$$

In the equation, *m*—Bragg diffraction order, *λ_max_*—diffraction wavelength, *d_hkl_*—interplanar spacing, *n_avg_*—average effective refractive index, and *θ*—incident angle of incident light.

Among them, the maximum reflected wavelength of the structural color is closely related to the crystal planar spacing of the photonic crystal, the refractive index of the material, and the angle of incidence of the light. Therefore, the photonic band gap can be effectively adjusted by changing the periodic structure in the photonic crystal, the refractive index of the material, and the grain size of the photonic crystal, as shown in Figure 5.

As Cu_2_O diameters were increased from 165 nm to 210 nm, 240 nm, and 275 nm, the color of the colored PET fabric changed from blue to green, yellow, and orange. The structural color diagram is shown in Figure 6.

The photographs and reflectance of the structural colored PET fabric are shown in Figure 7. As can be seen from Figure 7, during the synthesis of Cu_2_O, four kinds of Cu_2_O microspheres with different diameters can be obtained by adjusting the ratio of citrate and Cu^2+^.

### 3.4. Colorfastness of Cu_2_O Structural Colored Fabric

In order to further confirm the color fastness of the prepared polyester fabric with structural color, friction fastness, fast light fastness, washing fastness, and bending tests were carried out. The friction test was carried out in accordance with the EU standard ISO 105-X12:2001 [34], and a friction color fastness tester was used to test the friction fastness of the cloth sample. As shown in Figure 8, after the friction test, there was no obvious scratching on the surface of the cloth sample, and the color remained unchanged. The friction grade of the fabric was certified to be level 5, which indicates that the friction resistance of the fabric had reached the standard of daily use.

According to the national standard GB/T 8427-2019 [35], the fastness to light of the cloth sample was 6, as shown in Figure 9, which proves that the prepared polyester with structural color on the surface has excellent fading resistance.

No matter whether after high-speed rotary washing or rapid water washing, the polyester fabric samples with structural color on the surface did not fade, indicating that the prepared polyester fabric with structural color on the surface also has good washable resistance, as shown in Figure 10.

After stretching and bending 10 times, the color of the polyester fabric sample with structural color on its surface was not damaged except for some folds, as shown in Figure 11, indicating that the prepared polyester fabric with structural color on its surface has high fastness.

### 3.5. Antibacterial Properties of Cu_2_O Structural Colored Fabric

The antibacterial properties of structural colored fabrics are shown in Figure 12. The initial concentration of bacterial suspension was 105 CFU/mL. In order to ensure the activity and accuracy of the bacterial suspension, two blank control plates were prepared for culturing, and the difference in colony number between the two parallel plates did not exceed 15%, indicating that the bacterial suspension activity was good and the test data were effective. After 10^−5^ dilution of the freshly prepared suspension, the plate was coated, and after 24 h culturing, the number of colonies on the plate was as shown in Figure 12. The viable bacterial concentration of the suspension can be calculated by the number of colonies produced on the Petri dish, as indicated in Figure 12. The concentration of *E. coli* colonies in bacterial culture dishes was 1.08 × 10^8^ CFU/mL, as shown in Figure 12A, and the number of *S. aureus* colonies was 8.7 × 10^7^ CFU/mL, as shown in Figure 12D. Bacterial growth was inhibited since the Cu_2_O nanoparticle reagent was added, as shown in Figure 12B,E. The original fabrics were less effective against *E. coli* and *S. aureus*. While the Cu_2_O structural colored fabrics showed an excellent antibacterial effect, both *E. coli* and *S. aureus* were almost completely killed after treatment by a Cu_2_O structural colored fabric. The amount of bacterial solution added on each Petri dish was 0.1 mL, so the actual number of colonies on each plate needs to be multiplied by 10. After five washes, the actual colony count had increased slightly, from 0 to 70 for *E. coli* and from 0 to 40 for *S. aureus*. The bacteriostasis rate was calculated according to formula 1, as shown in Table 2.

As can be seen from the data in Table 2, the antibacterial samples had bacteriostatic rates ≥90% against Gram-negative *E. coli* and Gram-positive *S. aureus* before and after washing, so these coating fabrics had antibacterial properties. The sterilization principle of Cu_2_O is mainly through the release of copper ions. These copper ions can interact with -SH, -N_2_H, -COOH, -OH, and other groups in the proteins of bacteria or fungi, resulting in the death of bacteria. In addition, positively charged Cu_2_O microspheres and negatively charged bacteria produce contact of Cu_2_O microspheres with bacteria through charge attraction, and then the Cu_2_Omicrospheres enter the cell of the bacteria, causing the cell wall of the bacteria to break and the cell fluid to drain, resulting in the death of the bacteria. At the same time, the Cu_2_Omicrospheres entering the cell can interact with the protein enzymes in the bacterial cells, leading the enzymes to become denatured and inactivated, so as to kill the bacteria [37].

## 4. Conclusions

In this study, we prepared a novel structural colored functional textile by mixing Cu_2_O microspheres and PDMS on polyester fabric. In the preparation process, the molar ratio of copper acetate to sodium citrate was adjusted to produce different-sized Cu_2_O single-crystal nanoballs. Successfully, four diameters of Cu_2_O microspheres (165 nm, 210 nm, 240 nm, and 275 nm) were obtained on PET fabric, resulting in blue, green, yellow, and orange structural colors. Due to the strong adhesion of PDMS, the Cu_2_O structural colored textile has excellent washing, friction, and bending fastness. The Cu_2_O structural colored textile also has a good antibacterial performance against Gram-positive Staphylococcus aureus and Gram-negative *Escherichia coli*. The bactericidal rates of a Cu_2_O structural colored textile before washing for *E. coli* and *S. aureus* can reach 99.90%. After washing, the bactericidal rates of a Cu_2_O structural colored textile for *E. coli* and *S. aureus* can reach 92.40% and 94.53%, respectively. Therefore, this study provides experimental evidence of the potential for developing vibrant, highly antibacterial, structural colored functional textiles.

## Figures and Tables

**Figure 1 nanomaterials-14-01478-f001:**
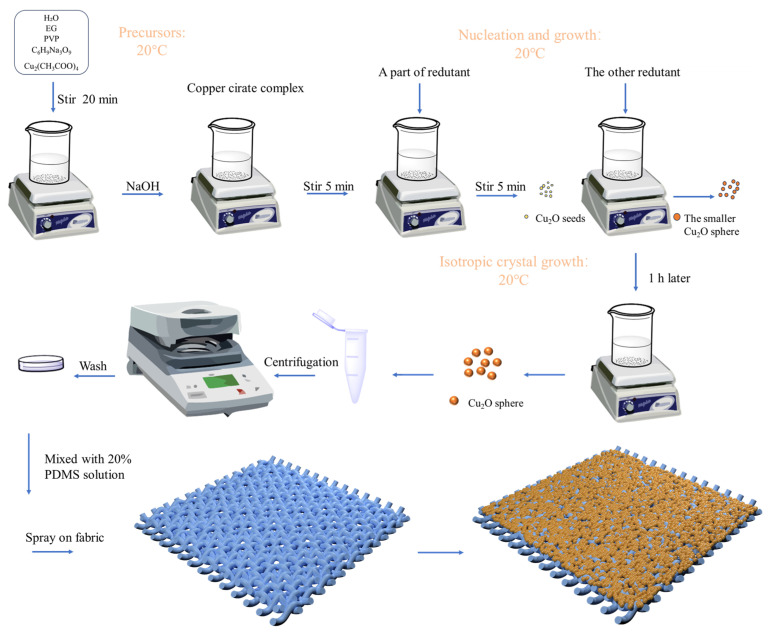
Preparation of Cu_2_O microspheres and simulation effect of Cu_2_O microspheres on fabrics.

**Figure 2 nanomaterials-14-01478-f002:**
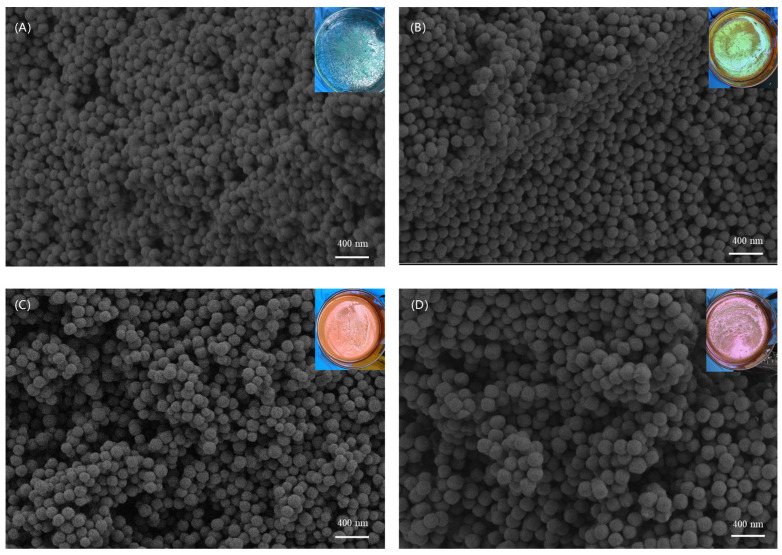
SEM images corresponding to Cu_2_O microspheres with different particle sizes: (**A**) 165 nm, (**B**) 210 nm, (**C**) 240 nm, (**D**) 275 nm.

**Figure 3 nanomaterials-14-01478-f003:**
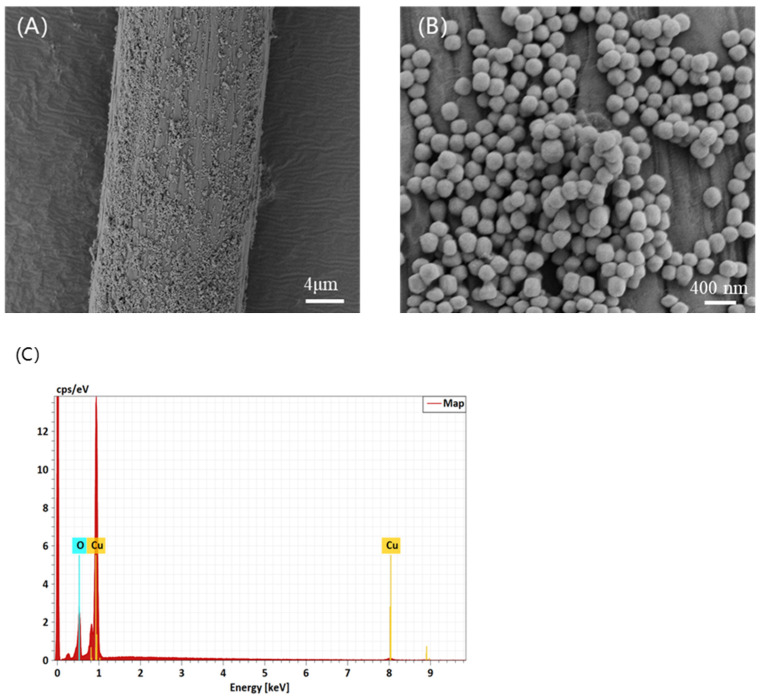
SEM images of (**A**) fiber surface sprayed with 210 nm Cu_2_O microspheres (×2000); (**B**) fabric surface sprayed with 210 nm Cu_2_O microspheres (×20,000); (**C**) EDS elemental analysis.

**Figure 4 nanomaterials-14-01478-f004:**
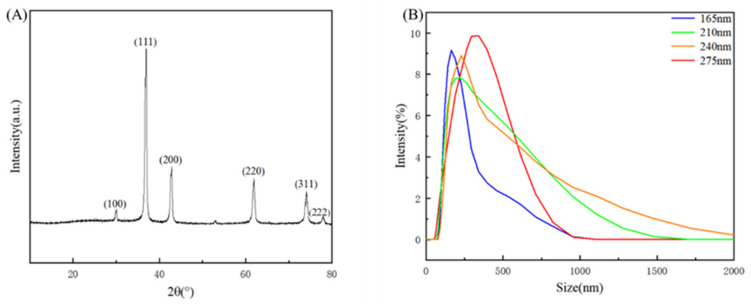
(**A**) XRD profiles of Cu_2_O microspheres. (**B**) Size distribution by intensity.

**Figure 5 nanomaterials-14-01478-f005:**
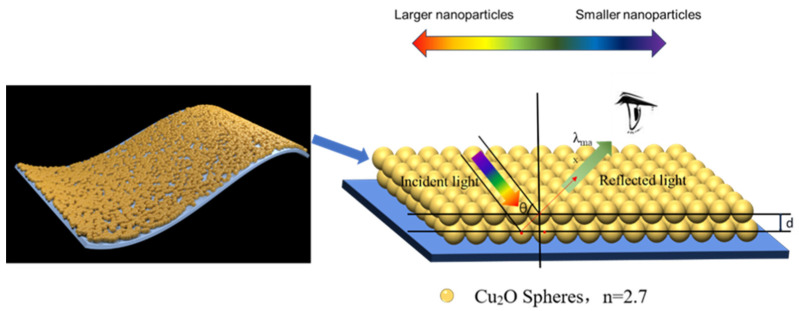
Structural color schematic.

**Figure 6 nanomaterials-14-01478-f006:**
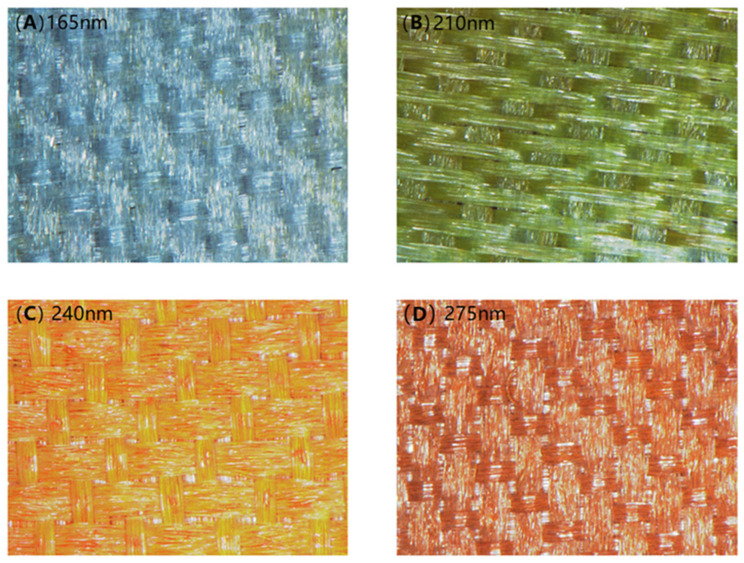
Photographs of structural colored fabrics: (**A**) 165 nm, (**B**) 210 nm, (**C**) 240 nm, (**D**) 275 nm.

**Figure 7 nanomaterials-14-01478-f007:**
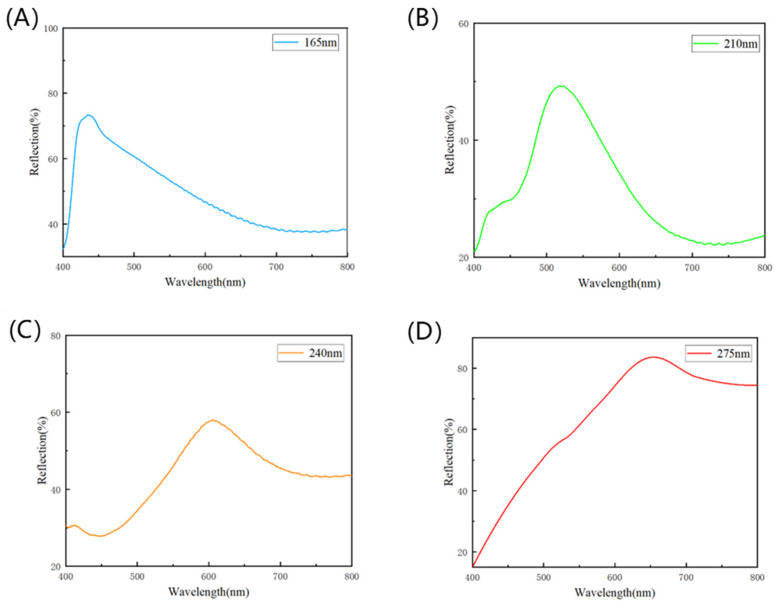
Reflectance curves of structural colored fabrics with different particle sizes of Cu_2_O microspheres: (**A**) 165 nm, (**B**) 210 nm, (**C**) 240 nm, (**D**) 275 nm.

**Figure 8 nanomaterials-14-01478-f008:**
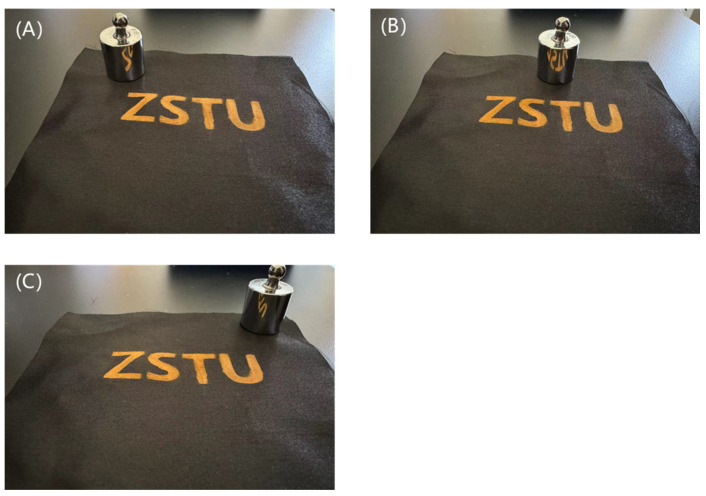
Comparison after rubbing fastness tests: (**A**) original fabric, (**B**) fabric in friction, (**C**) fabric after friction.

**Figure 9 nanomaterials-14-01478-f009:**
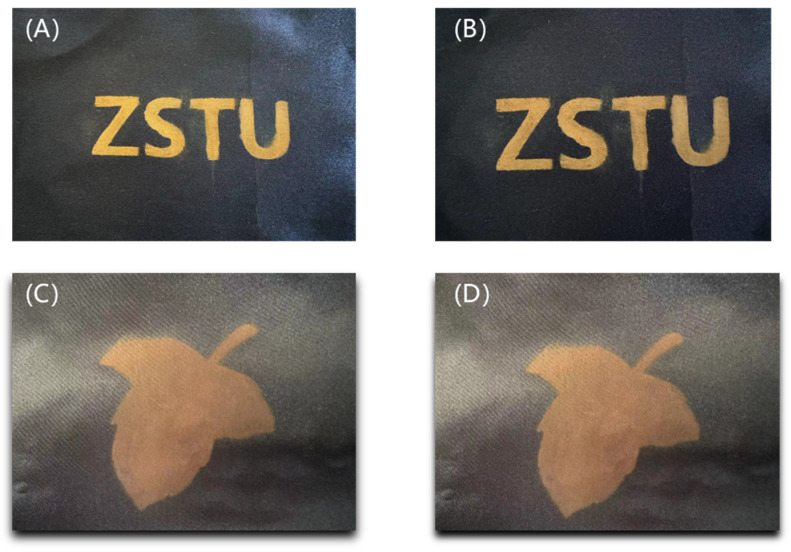
Comparison after fast light fastness tests: (**A**) original fabric, (**B**) fabric after fast light fastness tests, (**C**) original fabric, (**D**) fabric after fast light fastness tests.

**Figure 10 nanomaterials-14-01478-f010:**
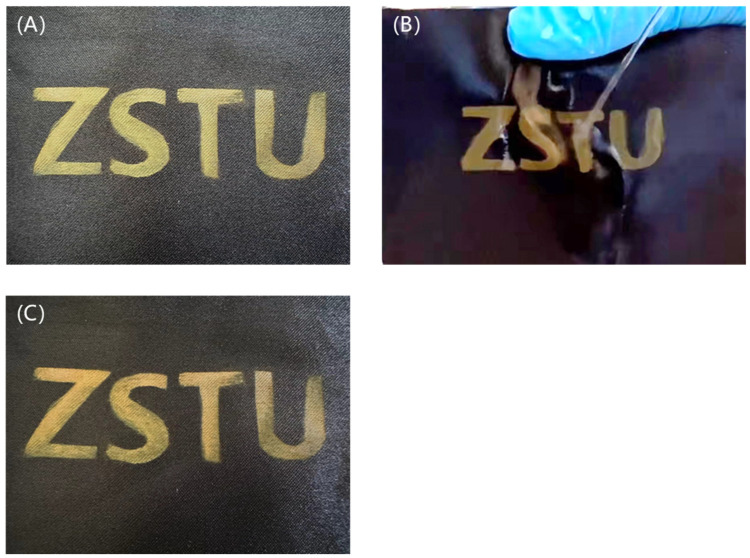
Comparison after washing fastness tests: (**A**) original fabric, (**B**) fabric in washing fastness tests, (**C**) fabric after washing fastness tests.

**Figure 11 nanomaterials-14-01478-f011:**
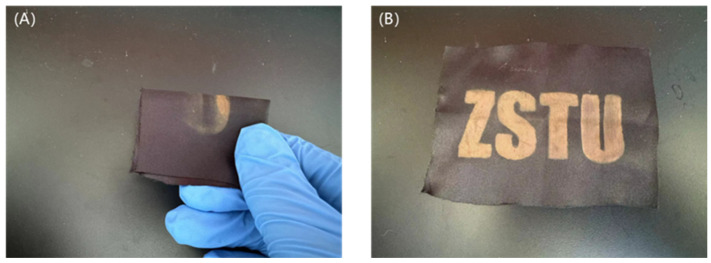
Comparison after rubbing bending tests: (**A**) fabric in rubbing bending tests, (**B**) fabric after rubbing bending tests.

**Figure 12 nanomaterials-14-01478-f012:**
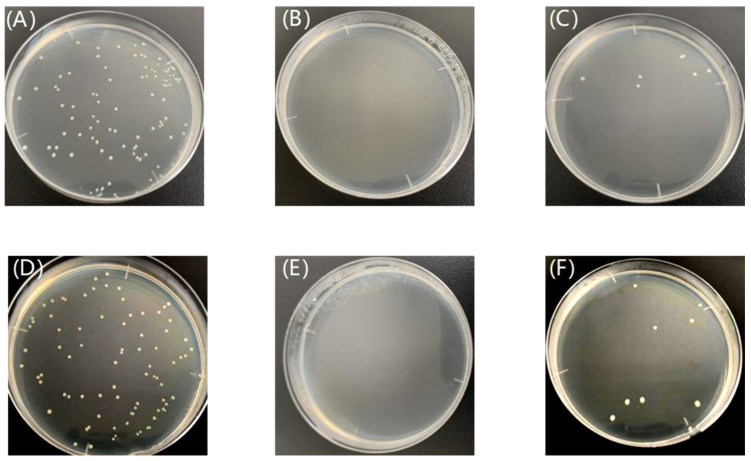
*E. coli* colony (**A**), control plate (**B**), and Cu_2_O structural colored fabric (**C**) after washes; *S. aureus* colony (**D**), control plate (**E**), and Cu_2_O structural colored fabric (**F**) after washes.

**Table 1 nanomaterials-14-01478-t001:** Composition table of chemical elements.

Element	Atomic Number	Percentage of Elements (%)
Cu	29	89.50
O	8	10.50
		100.00

**Table 2 nanomaterials-14-01478-t002:** Bacteriostatic rates of structural color fabric.

Sample	Inhibition Rates of *S. aureus*/%				Inhibition Rates of *E. coli*/%			
	Before Washing	CV	After Washing	CV	Before Washing	CV	After Washing	CV
Untreated	/	/	/	/	/	/	/	/
Cu_2_O structural fabric	99.90%	0.5%	92.40%	4.5%	99.90%	0.5%	94.53%	5.0%

## Data Availability

All data used in this study are available upon request from the corresponding author.

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
