# Peer review of "Construction of Patterned Cu2O Photonic Crystals on Textile Substrates for Environmental Dyeing with Excellent Antibacterial Properties"

_nanomaterials, 2024, doi:10.3390/nano14181478_

Round 1

Reviewer 1 Report

Comments and Suggestions for Authors

Dear Authors,

Thank you for the opportunity to review this manuscript. Since my expertise is microbiology, I have mostly reviewed this part, and it has many flaws that need to be modified. My comments are in the PDF file. Also, English language needs to be thoroughly checked, as some sentences make no sense.

Comments on the Quality of English Language

Even though I am not a native English speaker, some sentences make no sense to me, and I can say that the language needs to be checked very seriously.

Author Response

Comments 1:”E. coli and S. aureus” Please use here name of the bacteria, and please use Italics for the bacterial names

Response 1:Thank you for pointing this out. We used the full bacterial name in the abstract section which can be found in page 1, lines 22-24. And revised the bacterial names in italics throughout the text .

Lines 22-24:Cu2O structured-color fabrics demonstrate excellent antimicrobial properties against bacteria such as Escherichia Coli(E. coli)and Staphylococcus aureus(S. aureus).

Comments 2: “Compared to conventional fabric dyeing technologies, structural colors offer significant advantages such as being eco-friendly, vibrant in color, and resistant to fading”This sentence is already written above.

Response 2:Thanks to the reviewer for this careful comment. We have crossed out this sentence in the article.

Comments 3:”Applying structural colors to fabric dyeing could fundamentally address the challenges associated with traditional methods and establish a novel approach to textile coloring. If the structural color is applied to fabric coloring, it is hoped to fundamentally solve the problems of conventional fabric coloring and become a new textile coloring method” This paragraph makes no sense, you are just repeating few phrases without some concrete information.

Response 3: We thank the reviewer for the positive and constructive comments regarding our paper. In the revised manuscript, the correspondent context has been reorganized, which can be found in page 2,lines 46-59.

Lines 46-59:The dyeing process for coloring fabrics is becoming more and more mature, mainly including dip-dyeing, tie-dyeing, cold-dyeing, printing and dyeing, etc., and the products obtained can cover all aspects of people's lives. However, there are still some problems under the background of advocating green and sustainable development: (1) In the dyeing process, you need to add dyes and a variety of chemicals and additives; processing technology is cumbersome and complex; energy consumption and water consumption are high; (2) Each process will produce a large number of difficult to deal with colored wastewater; and wastewater composition is complex, with high treatment costs that are difficult to effectively remove; (3) It is easy to pollute the environment. The structure of the pigment molecule itself is complicated, containing several aromatic rings bonded into a fully conjugated system with a long series of alternating single and double bonds between carbon atoms and other atoms, which makes it difficult to be synthesized artificially. In addition, the structure of the pigment molecule itself is not stable, and it is easy to be decomposed under long-term sunshine.

Comments 4:”Gram-positive and Gram-negative”Please do not capitalize letter G when writing gram positive or negative.

Response 4: Thank you for kindly reminding us. In manuscripts with “Gram positive” or “negative,” the letter G has been changed to lowercase.

Comments 5:”method”Please provide all the details so the experiment can be replicated. If you just mention the standards, someone that does not have, or does not want to buy the standard cannot replicate the method. Please write exactly the method protocol, what was the bacterial concentration, what media etc... thank you

Response 5: We thank the reviewer for raising these important points. We have followed the reviewer's request write exactly the method protocol, what was the bacterial concentration, what media etc,which can be found in page 5,section 2.6. Antimicrobial Testing,page5,lines 185-206.

Lines 185-206:Antimicrobial performance: According to GB/T 20944.3-2008 “Evaluation of Antimicrobial Performance of Textiles Part 3: Oscillation Method”, E.coli and S.aureus are the test strains. The source of the bacteria is purchased from Beijing SanYao Science&Technology Co.,Ltd. The specific test procedure was as follows:

The samples to be examined were cut into pieces of 5 mm × 5 mm size, weighed 0.75 g, and sterilized by pressure steam at 103 kPa and 125 °C for use. The control sample was treated in the same way. Triangular vials were filled with control sample and treated sample, then 70 mL of 0.03 mol/L phosphate (PBS) buffer was added to each triangular vial. Add 5 mL of bacterial suspension into the above triangular vials and place them on a thermostatic shaker at 150 r/min and 25 ℃ for 18 h. After 18h of contact shaking, 1mL of liquid was sucked from each flask and added into 9 mL of 0.03 mol/L PBS buffer, mixed thoroughly, and then diluted to the appropriate dilution gradient by 10-fold dilution method, and then 0.1 mL was sucked from each dilution gradient and added into agar medium, and two parallel samples were made for each dilution, and then inverted after coating evenly, and then continuously cultured at 37 ℃ for 24h in a thermostatic incubator. The sample was incubated at 37 ℃ for 24 h, and the number of colonies on each medium was photographed, counted and recorded.

Calculate the inhibition rate of the sample to be tested against the target strain:

(1)

Y--Inhibition rate of the sample to be tested (%);

Wb--concentration of live bacteria after 18 h of the blank sample (CFU/mL);

Wc--concentration of live bacteria after 18 h of the sample to be tested (CFU/mL).

Comments 6: “GB/T 20944.3-2008”Please give more detail and the reference where this method could be found.

Response 6:Thanks for your professional suggestions,references to this approach have been cited in section 2.6 of the manuscript.

Comments 7:”S.Ureu”This is spelled wrong, please write S. aureus , and in italics, both the full name and abbreviation. Also name what was the source of your bacteria, was it purchased, isolated (how), or how was it obtained...

Response 7: We thank the reviewer for the positive feedback. I have fixed the error in the manuscript "S.Ureu" and changed it to italics. And the source of the bacteria is purchased from Beijing SanYao Science&Technology Co.,Ltd.

Comments 8:In figure 12,Both E coli and S aureus look the same to me, are you sure these are the correct plates? Also, what does "parallel plate" mean, is it the control, meaning fabric that was not dyed? please specify this.

Response 8:Thanks for pointing out the error. We have changed "parallel plate" to "control plate",which can be found in page 12,lines 314-317.

Lines 314-317:Figure 12. Figure of the E.coli colony (A) control plate (B) Cu2O structural colored fabric (C) after washes; figure of the S. aureus colony (D) control plate (E) Cu2O structural colored fabric (F) after washes

Comments 9:The antibacterial properties of structure-colored fabrics are shown in Figure 12. As shown in Figure 12, for control sample, the number of E.coli colonies in bacterial culture dishes is 108, as shown in Figure 12 (A), the number of S.Ureu colonies is 83, as shown in Figure 12(D). Bacterial growth was inhibited since the Cu2O nanoparticle reagent was added as shown in Figure 12(B),(E). The original fabrics show less effective against E. coli and S. aureus. While the Cu2O structural colored fabrics show excellent antibacterial effect, both E. coli and S. aureus are almost completely killed after treatment by Cu2O structural colored fabric. After washing, the colonies just increased slightly, from 0 to 7 for E.coli and from 0 to 4 for S.Ureu. The bacteriostasis rate was calculated according to the standard as shown in Table 2. There are 5 problems in this paragraph.

Response 9: We sincerely thanks you for your feedback which would help to improve the quality of our manuscript. We have modified the paragraph,which can be find on page 12,lines 318-335.

Lines 318-335:The concentration of bacterial suspension was 105 CFU/mL. In order to ensure the activity and accuracy of the bacterial suspension, two blank control plates were prepared for culture, and the difference of colony number between the two parallel plates did not exceed 15%, indicating that the bacterial suspension activity was good and the test data were effective. After 10-5 dilution of the freshly prepared suspension, the plate was coated, and after 24 h culture, the number of colonies on the plate was shown in Figure 12. The viable bacteria concentration of the suspension can be calculated by the number of colonies produced on the petri dish in Figure 12. The concentration of E.coli colonies in bacterial culture dishes is 1.08*10CFU/mL, as shown in Figure12(A), the number of S. aureus colonies is 8.3*107 CFU/mL, as shown in Figure 12(D). Bacterial growth was inhibited since the Cu2O nanoparticle reagent was added as shown in Figure 12(B),(E). The original fabrics show less effective against the E. coli and S. aureus. While the Cu2O structural colored fabrics show excellent antibacterial effect, both E. coli and S. aureus are almost completely killed after treatment by Cu2O structural colored fabric. The amount of bacterial solution added on each petri dish is 0.1 mL, so the actual number of colonies on each plate needs to be ×10. After 5 times wash, the actual colony count just increased slightly, from 0 to 70 for E.coli and from 0 to 40 for S. aureus. The bacteriostasis rate was calculated according to the formula 1 as shown in Table 2.

Comments 10:”gram-negative E.coli and Gram-positive S.Ureu” There are spelling errors.

Response 10:Thank you for pointing this spelling errors. We found and corrected these spelling errors in the manuscript.

Comments 11:Rewrite” In addition, positively charged copper nanoparticles and negatively charged bacteria, through the charge attraction makes the copper nanoparticles contact with the fine and so on, and then the copper nanoparticles nter into the cells of the bacteria, which makes the cell wall of the bacteria break through the wall, and the cytosol outflow, resulting in the death of the bacteria. At the same time, the copper nanoparticles can interact with the protein enzymes in the bacterial cells, which can denature and inactivate the enzymes, thus killing the bacteria.”

Response 11: We would like to thank you for your professional review work, constructive comments, and valuable suggestions on our manuscript. We have rewritten the paragraph, which can be found on page 13,lines 345-354.

Lines 345-354:The sterilization principle of Cu2O is mainly through the release of copper ions. These copper ions can act with -SH, -N2H, -COOH, -OH and other groups in the proteins of bacteria or fungi, resulting in the death of bacteria. In addition, positively charged copper nanoparticles and negatively charged bacteria contact copper nanoparticles with bacteria through charge attraction, and then the copper nanoparticles enter the cell of the bacteria, causing the cell wall of the bacteria to break, and the cell fluid to drain, resulting in the death of the bacteria. At the same time, the copper nanoparticles entering the cell can interact with the protein enzymes in the bacterial cells, making the enzymes denaturated and inactivated, so as to kill the bacteria.

Reviewer 2 Report

Comments and Suggestions for Authors

Your manuscript entitled “Construction of patterned Cu2O photonic crystals on textile substrates with outstanding washing fastness and excellent an-tibacterial properties” is interesting for a publication in Nanomaterials as it highlights the use of Cu2O microspheres for creating eco-friendly, vibrant, and durable structural colors on fabrics. These microspheres, applied via a spraying method, produce colors with high resistance to fading and enhance fabric properties like hydrophobicity and antimicrobial activity. Unlike traditional dyes, this approach avoids environmental harm and addresses challenges like colorfastness and breathability in fabric coloring. The study also explores the antibacterial effectiveness of these fabrics, providing a foundation for next-generation textiles that combine aesthetic appeal with enhanced performance and sustainability. However, I found that this manuscript requires major revision prior the acceptance in Nanomaterials.

1.     From the Materials and Methods, How does the variation in Cu2O microsphere particle size (165-275 nm) influence the structural color and washing fastness of the coated fabric, and are there any trade-offs between particle size and the uniformity or durability of the color? Considering the spray method used for Cu2O-PDMS coating, how does the nozzle diameter (0.2 mm) and compressor pressure (0.5 PSI) impact the evenness of the coating and the overall performance of the structural colored fabric?

2.     In Figure 2,  How does the SEM analysis ensure that the observed uniformity of Cu2O microsphere distribution and size on the fabric surface correlates with the washing fastness and antibacterial properties? Could variations in microsphere distribution at different scales affect these functional properties?

3.     How does the preferential orientation of the (111) and (200) planes in the XRD analysis of Cu2O microspheres of Figure 4 influence their optical properties and functionality on textile substrates? Could this crystallographic orientation affect the consistency of structural color and the antibacterial performance across different fabric samples?

4.     How does the variation in Cu2O microsphere diameter impact the uniformity and stability of the structural color across the fabric as in Figure 5, especially considering potential challenges with consistent periodicity and refractive index in practical textile applications? Could slight deviations in microsphere size or arrangement affect color accuracy and durability?

5. Potential of this photonic crystal structure as biosensors and virus detections should be highlighted in the introduction or discussion. Please read and cite this review Tsalsabila, A., Advanced Optical Materials 2400849 2024.

Comments on the Quality of English Language

The title is little bit confusing. Consider to cut some words or paraphrase it.

Author Response

Comments 1: 1.From the Materials and Methods, How does the variation in Cu2O microsphere particle size (165-275 nm) influence the structural color and washing fastness of the coated fabric, and are there any trade-offs between particle size and the uniformity or durability of the color? Considering the spray method used for Cu2O-PDMS coating, how does the nozzle diameter (0.2 mm) and compressor pressure (0.5 PSI) impact the evenness of the coating and the overall performance of the structural colored fabric?

Response 1:We would like to thank you for your professional review work, constructive comments, and valuable suggestions on our manuscript. The human eye will be able to see the vibrant structural colors if the wavelength is in the visible region. Among them, the maximum reflected wavelength of the structural color is closely related to the particle size of the nanosized microspheres, the refractive index of the material, and the angle of incidence of light. Therefore, the photonic band gap can be effectively adjusted by changing the particle size in the nanospheres as well as changing the refractive index of the material. The washing fastness of microspheres of different particle sizes sprayed onto fabrics is the same. Most of the nano-microspheres on the textile substrate to build the structure of the color and the textile substrate between the bonding fastness is poor, to improve the fastness usually need to be added to the system of the binder. Variations in nozzle diameter and compressor pressure affect the range and speed of spraying, and we have chosen these parameters because of the small size of the fabrics selected. The effect on the uniformity of the coating and the overall performance of the structural color fabrics is minimal.

  1. In Figure 2, How does the SEM analysis ensure that the observed uniformity of Cu2O microsphere distribution and size on the fabric surface correlates with the washing fastness and antibacterial properties? Could variations in microsphere distribution at different scales affect these functional properties?

Response 2:We sincerely thank you for your feedback which would help to improve the quality of our manuscript. We usually use SEM to analyze the morphology of microspheres, such as observing whether the microsphere particles form regular spheres, and the corresponding colors of microspheres with different particle sizes. Changes in the distribution of microspheres at various scales can cause color changes, but have little effect on wash fastness and antimicrobial properties. Antimicrobial properties are tested according to GB/T 20944.3-2008 “Evaluation of Antimi-crobial Performance of Textiles Part 3: Oscillation Method”. Simulated washing test: the fabric was put into a 150 mL beaker filled with water, magnetically stirred at 1000 r/min for 15 min. And the SEM testing is only to test the surface structure.

  1. How does the preferential orientation of the (111) and (200) planes in the XRD analysis of Cu2O microspheres of Figure 4 influence their optical properties and functionality on textile substrates? Could this crystallographic orientation affect the consistency of structural color and the antibacterial performance across different fabric samples?

Response 3:We are grateful to the reviewer for bringing up these crucial points. In this paper, the growth rate of crystals along the (111) and (200) directions is controlled by modifying the pH value of the solution and the concentration of the solution such as ethylene glycol. Moreover, the growth rate of the crystal faces is regulated by PVP through reducing the surface energy of the crystal faces, which prompts the crystals to grow along the directions of several main axes simultaneously and eventually forms a structure similar to a sphere. By adjusting the experimental parameters, the preparation of spherical Cu2O crystals has been accomplished. Unlike the previous asymmetric scattering films that demand the combination of noble metal nanoparticles and high refractive index films, and have certain requirements for the thickness of the high refractive index films, the Cu2O film with structural color merely requires one or two layers of single crystal spheres, which can support the obvious asymmetric scattering phenomenon in a thin situation.

  1. How does the variation in Cu2O microsphere diameter impact the uniformity and stability of the structural color across the fabric as in Figure 5, especially considering potential challenges with consistent periodicity and refractive index in practical textile applications? Could slight deviations in microsphere size or arrangement affect color accuracy and durability?

Response 4:We thank the reviewer for the positive and constructive comments regarding our paper. The variation in microsphere diameter leads to different color rendering effects on the fabric surface, ranging from blue at short wavelengths to red at long wavelengths. The mechanism is explained in Figure 5. The uniformity of color rendering is determined by the distribution effect of the microspheres, and they are better distributed on the fabric surface in this article. The bumps and folds on the textile surface do not affect the color rendering effect of the microspheres but have an impact on the luster effect. If the microspheres are distributed on a smooth surface, such as the glassware in Figure 2, the overall gloss is superior. The change in microsphere size has a significant effect on the color, as described in the paper, but the arrangement of microspheres has no obvious influence on the color of the fabric.

  1. Potential of this photonic crystal structure as biosensors and virus detections should be highlighted in the introduction or discussion. Please read and cite this review Tsalsabila, A., Advanced Optical Materials 2400849 2024.5.

Response 5:We would like to thank the reviewer for these constructive comments. We have emphasized this in the Introduction section of the manuscript and cited the article, which can be found on page 1, lines 40-43.

Lines 40-43:Due to the unique properties of these nanostructures, it is possible to achieve specific functionalities such as antimicrobial properties, hydrophobicity, self-cleaning capabilities, UV resistance and as a photonic biosensor to detect viruses[4–6].

Reviewer 3 Report

Comments and Suggestions for Authors

In this manuscript, the feasibility of utilizing copper (I) oxide nanoparticles as a coloring agent with exceptional resilience to mechanical forces and notable antimicrobial characteristics is investigated.It is indubitable that this study is of considerable practical interest. Nevertheless, it is important to acknowledge that the work leaves a somewhat controversial impression due to the presence of numerous significant errors and inaccuracies.

A thorough examination of the manuscript reveals the necessity for significant revisions.

General:

1. The manuscript file should be updated with line numbers.

2. The text of the manuscript, including the introduction, contains a substantial amount of information regarding the utilization of copper oxide nanomaterials as dyes for fabric. Was the primary objective to obtain and utilize copper oxide nanoparticles as dyes for fabric? If the answer is in the affirmative, the authors are encouraged to consider a title change.

3. All Latin names should be italicized. Please verify this.

Materials and Methods:

4. In the section entitled " 2.2 Preparation of single-crystal Cu2O microsphere," the authors indicated that copper oxide nanoparticles with a monomodal distribution, measuring between 165 and 275 nanometers, were obtained, which represents the result. However, the subsequent measurement data are absent. Additionally, there is no description of how the nanoparticles were further sorted into separate fractions or how four kinds of Cu2O nanoparticle samples with different size distributions were obtained.

5. Section 2.3 Preparation of Cu2O structure colored fabric. The text includes an unclear reference to a "spraying method." It is necessary to provide the manufacturer of the 184 silicone elastomer base material and the 184 silicone elastomer curing agent.

6. The section entitled "2.4 Characterization Methods" contains a plethora of information that would be more readily comprehensible to the reader if it were divided into discrete paragraphs. A comprehensive description of the physical methods employed to characterize the resulting materials is essential.

7. Furthermore, the section "2.4 Characterization Methods" would be better organized if the antibacterial studies were placed in a separate section. The microbiological method employed to assess the antibacterial efficacy of the materials obtained should be delineated in comprehensive detail. Additionally, the information pertaining to the nutrient medium (agar) utilized for bacterial cultivation, the incubation temperature, and the duration of the incubation period should be incorporated. The microbiological methodology should be described in sufficient detail to allow for replication by other researchers, include appropriate controls, and be written in a way that is accessible to the reader.

8. In section "3.1. SEM images of Cu2O structural colored fabric", it is stated that four different samples of Cu2O NPs were obtained under different conditions. However, there is no information about this in the materials and methods.

9. The species of Staphylococcus aureus is incorrectly stated as "S.Ureu," p. 12. The correct spelling is "S. aureus."

10. Indicate which is the control in Figure 12.

Author Response

  1. The manuscript file should be updated with lines numbers.1.

Response 1:Thanks for pointing out our problems. Lines numbers have been added to the manuscript file.

  1. The text of the manuscript, including the introduction, contains a substantial amount of information regarding the utilization of copper oxide nanomaterials as dyes for fabric. Was the primary objective to obtain and utilize copper oxide nanoparticles as dyes for fabric? If the answer is in the affirmative, the authors are encouraged to consider a title change.2.

Response 2:We thank the reviewer for this insightful comment. We have changed the title based on the content of the article. The revised title is:Construction of patterned Cu2O photonic crystals on textile substrates for environmental dying with excellent antibacterial properties.

  1. All Latin names should be italicized. Please verify this.

Response 3:Thank you for pointing this out. All Latin names have been italicized in the manuscript.

Materials and Methods: 

  1. In the section entitled " 2.2 Preparation of single-crystal Cu2O microsphere," the authors indicated that copper oxide nanoparticles with a monomodal distribution, measuring between 165 and 275 nanometers, were obtained, which represents the result. However, the subsequent measurement data are absent. Additionally, there is no description of how the nanoparticles were further sorted into separate fractions or how four kinds of Cu2O nanoparticle samples with different size distributions were obtained.4.

Response 4:We sincerely thanks you for your feedback which would help to improve the quality of our manuscript. In 2.2. Preparation of Cu2O microsphere, we added that the amount of other reagents remained the same and by varying the ratios of copper acetate and trisodium citrate (0.8, 1.0, 1.1, and 1.2), Cu2O microspheres with different particle sizes could be obtained. The particles with different particle sizes were prepared in batches and not prepared and sorted together. As can be seen in Fig. 4.(B), the nanoparticles prepared in the same ratio have a more concentrated particle size. We do not need to categorize the different compositions. We made changes to the paragraph, which can be found on page 3, lines 146-150.

Lines 146-150: By varying the ratios of copper acetate and trisodium citrate (0.8, 1.0, 1.1, and 1.2), Cu2O nano-microspheres with corresponding particle sizes of 165 nm, 210 nm, 240 nm, and 275 nm were finally prepared. The diameters of Cu2O microspheres prepared in different ratios are different with a few nanometers deviation in the particle size. The preparation process is shown in Figure 1.

  1. Section 2.3 Preparation of Cu2O structure colored fabric. The text includes an unclear reference to a "spraying method." It is necessary to provide the manufacturer of the 184 silicone elastomer base material and the 184 silicone elastomer curing agent.5.

Response 5:We thank the reviewer for raising these important points.The spraying method mentioned in Section 2.3 of Part 2 has been modified to make the description more accurate which can be found on page 4, lines 160-161. Manufacturer information is described in 2.1.materials. It can be found on page 3, lines 130-132.

Lines 160-161:A mixed solution of PDMS and Cu2O is sprayed onto the surface of polyester fabric using a spray gun connected to an air compressor. The compressor pressure was set to 0.5 PSI, and the spray gun nozzle diameter was set to 0.2 mm. The Cu2O-PDMS coated fabric was dried at 50°C for 3 hours to obtain the structural colored fabric.

Lines 130-132:SYLGAR 184 Silicone Elastomer(PDMS) was purchased from Dow Corning Co., Ltd.

  1. The section entitled "2.4 Characterization Methods" contains a plethora of information that would be more readily comprehensible to the reader if it were divided into discrete paragraphs. A comprehensive description of the physical methods employed to characterize the resulting materials is essential.

Response 6:We thank the reviewer for the positive and constructive comments regarding our paper. The section entitled “2.4 Characterization Methods” has been divided into separate paragraphs, and the physical methods used to characterize the resulting materials are described in detail.

  1. Furthermore, the section "2.4 Characterization Methods" would be better organized if the antibacterial studies were placed in a separate section. The microbiological method employed to assess the antibacterial efficacy of the materials obtained should be delineated in comprehensive detail. Additionally, the information pertaining to the nutrient medium (agar) utilized for bacterial cultivation, the incubation temperature, and the duration of the incubation period should be incorporated. The microbiological methodology should be described in sufficient detail to allow for replication by other researchers, include appropriate controls, and be written in a way that is accessible to the reader.7.

Response 7:We would like to thank you for your professional review work, constructive comments, and valuable suggestions on our manuscript. We have included antimicrobial studies as a separate section in section 2.5 of the article and have detailed the microbiological methods used to assess the antimicrobial efficacy of the obtained materials.

  1. In section "3.1. SEM images of Cu2O structural colored fabric", it is stated that four different samples of Cu2O NPs were obtained under different conditions. However, there is no information about this in the materials and methods.8.

Response 8:Thanks to the reviewer for this careful comment. About four different Cu2O nanoparticle samples were prepared under different conditions. In 2.2.Preparation of Cu2O microsphere it is described that the amount of other reagents remained the same and by varying the ratios of copper acetate and trisodium citrate (0.8, 1.0, 1.1, and 1.2), Cu2O microspheres with different particle sizes could be obtained. The Cu2O microspheres with different diameters were synthesized by altering the ratio of citrate to Cu2+. Upon increasing the ratio of citrate to Cu2+ from 0.8 to 1.2, the diameters of the Cu2O microspheres can be tuned from 165 nm to 275 nm. More sodium citrate leads to fewer free Cu2+ ions in the system, which reduces the number of nuclei in the nucleation stage and increases the size of the final growth products. We made changes to the paragraph, which can be found on page 3, lines 146-150.

Lines 146-150: By varying the ratios of copper acetate and trisodium citrate (0.8, 1.0, 1.1, and 1.2), Cu2O nano-microspheres with corresponding particle sizes of 165 nm, 210 nm, 240 nm, and 275 nm were finally prepared. The diameters of Cu2O microspheres prepared in different ratios are different with a few nanometers deviation in the particle size. The preparation process is shown in Figure 1.

  1. The species of Staphylococcus aureusis incorrectly stated as "S.Ureu," p. 12. The correct spelling is "S. aureus."

Response 9:Thanks for pointing out the error. This spelling error has been fixed in the full text.

  1. Indicate which is the control in Figure 12.

Response 10:Thank you for the helpful comments. Figures (A) and (D) in Figure 12 are the controls.

Round 2

Reviewer 1 Report

Comments and Suggestions for Authors

The article is fine after revision. just please make a detail spellcheck, in several spaces it is written S.aureus and E.coli, and there should be space between the words...

Comments on the Quality of English Language

The article is fine after revision. just please make a detail spellcheck, in several spaces it is written S.aureus and E.coli, and there should be space between the words...

Author Response

Comments 1: The article is fine after revision. just please make a detail spellcheck, in several spaces it is written S.aureus and E.coli, and there should be space between the words...

Response 1: Thank you for kindly reminding us. We have carefully checked the spelling and have revised S.aureus and E.coli to S. aureus and E. coli in the manuscript.

Reviewer 2 Report

Comments and Suggestions for Authors

Thanks for addressing my comments. I will recommend to be published as it is.

Comments on the Quality of English Language

The English is good enough. Minor check is necessary to make it shorter. For instance, the abstract still can be enhanced.

Author Response

Comments 1: The English is good enough. Minor check is necessary to make it shorter. For instance, the abstract still can be enhanced.

Response 1: We thank the reviewer for the positive and constructive comments regarding our paper. We have made revisions to the abstract section.

Reviewer 3 Report

Comments and Suggestions for Authors

The authors have performed an admirable job on the manuscript and have implemented the requisite revisions. Nevertheless, a number of significant comments and recommendations for improvement remain.

1. Please provide the exact initial concentration of bacterial cells (CFU/mL) in the description of the method for assessing antibacterial activity (line 185).

2. The authors' use of the term "microspheres" to describe nanoparticles that are in fact measured in nanometers is a confusing and inaccurate choice of words.

3. The manuscript contains numerous careless, spelling, and stylistic errors that the authors should carefully reread and correct. For example, the spelling of Cu2O.

4. Lines 337-346 lack any references to sources.

Author Response

Comments 1: Please provide the exact initial concentration of bacterial cells (CFU/mL) in the description of the method for assessing antibacterial activity (line 185).

Response 1: We sincerely thank you for your feedback which would help to improve the quality of our manuscript. The exact initial concentration of bacterial cells has been provided in the paper, which can be found in page 12, lines 315-316.

Lines 315-316:The initial concentration of bacterial suspension was 105 CFU/mL.

Comments 2: The authors' use of the term "microspheres" to describe nanoparticles that are in fact measured in nanometers is a confusing and inaccurate choice of words.

Response 2: We sincerely thank you for your feedback which would help to improve the quality of our manuscript. We replaced the words "nanoparticles" with "microspheres." The reason why "microspheres" are used to describe Cu2O nanomaterials is that the definition of nanometer should be below 100 nanometers in size.

Comments 3: The manuscript contains numerous careless, spelling, and stylistic errors that the authors should carefully reread and correct. For example, the spelling of Cu2O.

Response 3: Thanks for pointing out the error. The manuscript has been carefully reread and corrected for carelessness, spelling and stylistic errors.

Comments 4: Lines 337-346 lack any references to sources.

Response 4: We thank the reviewer for the positive and constructive comments regarding our paper. We have cited references in the article.